# Insights into Phytoplankton Dynamics and Water Quality Monitoring with the BIOFISH at the Elbe River, Germany

**Andre Wilhelms** [1,*] , **Nicolas Börsig** [1] , **Jingwei Yang** [2] , **Andreas Holbach** [3] and **Stefan Norra** [4]

1   Working Group Environmental Mineralogy and Environmental System Analysis (ENMINSA), Institute of Applied Geosciences, Karlsruhe Institute of Technology, Kaiserstraße, 12, 76131 Karlsruhe, Germany; nicolas.boersig@kit.edu

2   Guangdong Provincial Engineering and Technology Center for Water Affairs and Water Ecology, Shenzhen 518001, China; yangjw@swpdi.com

3   Department of Ecoscience, Aarhus University, Frederiksborgvej, 399, 4000 Roskilde, Denmark; anho@ecos.au.dk

4   Department of Soil Sciences and Geoecology, Institute of Environmental Sciences and Geography, Potsdam University, Campus Golm, Building, 12, 14476 Potsdam, Germany; stefan.norra@uni-potsdam.de

*   Correspondence: andre.wilhelms@kit.edu; Tel.: +49-721-608-44878

**Abstract:** Understanding the key factors influencing the water quality of large river systems forms an important basis for the assessment and protection of cross-regional ecosystems and the implementation of adapted water management concepts. However, identifying these factors requires in-depth comprehension of the unique environmental systems, which can only be achieved by detailed water quality monitoring. Within the scope of the joint science and sports event "Elbschwimmstaffel" (swimming relay on the river Elbe) in June/July 2017 organized by the German Ministry of Education and Research, water quality data were acquired along a 550 km long stretch of the Elbe River in Germany. During the survey, eight physiochemical water quality parameters were recorded in high spatial and temporal resolution with the BIOFISH multisensor system. Multivariate statistical methods were applied to identify and delineate processes influencing the water quality. The BIOFISH dataset revealed that phytoplankton activity has a major impact on the water quality of the Elbe River in the summer months. The results suggest that phytoplankton biomass constitutes a substantial proportion of the suspended particles and that photosynthetic activity of phytoplankton is closely related to significant temporal changes in pH and oxygen saturation. An evaluation of the BIOFISH data based on the combination of statistical analysis with weather and discharge data shows that the hydrological and meteorological history of the sampled water body was the main driver of phytoplankton dynamics. This study demonstrates the capacity of longitudinal river surveys with the BIOFISH or similar systems for water quality assessment, the identification of pollution sources and their utilization for online in situ monitoring of rivers.

**Keywords:** water quality; phytoplankton; river dynamics; multisensor system; online monitoring; high spatial resolution; multivariate statistics

## 1. Introduction

Near the end of the 1980s, the Elbe River was one of the most polluted rivers in Central Europe. The high pollution of the river and its sediments was caused by agriculture, industrial activities and inadequate or non-existent wastewater treatment in the German Democratic Republic (GDR) and Czechoslovakia [1,2]. Industrial activity in the Czech section of the Elbe included chemical and pharmaceutic industries as well as the pulp industry [3]. Major pollution sources in the German section of the Elbe were pharmaceutical and pulp industry in the Upper Elbe section and the tributaries Mulde and Saale in the middle section of the Elbe. The Mulde and the Saale, which drain large parts of the industrial region in Central Germany, were severely polluted with organic substances,

nutrients and trace metals by wastewater from chemical and salt mining industries in GDR times [4,5]. The water quality of the Elbe improved following German reunification in 1990. Collapse of many industrial and agricultural complexes in the former GDR and campaigns initiated by the International Commission for the Protection of the Elbe River (IKSE) resulted in a decrease of discharged pollutants. The improvement of water quality can also be attributed to the construction of numerous wastewater treatment plants [3,6,7]. Research projects and monitoring programs to investigate and evaluate the changes in water quality were launched by various institutions such as the ARGE ELBE, the IKSE and the Federal Ministry of Research and Technology (BMBF) [8]. Their results are well-documented in numerous reports [9–11]. Moreover, since 2001, an information system called "ELBIS" has offered public users insights into the development of the impact of pollution on Elbe sediments and water quality [12].

Several studies applied multivariate statistical methods to assess the water quality of the Elbe River. Pepelnik et al. [13] analyzed samples taken with high spatial resolution along the Elbe River over a period of four years. They used cluster analysis to distinguish between elements of different origins, such as geogenic- and anthropogenic-influenced elements. Petersen et al. [14] used principal component analysis to identify processes that involve the changes of multiple water parameters. They found two components related to discharge and biological activity, which were sufficient to explain most of the observed total variance in the dataset. The component for "biological activity" can be attributed to biomass being modified by biological processes. It exhibits parallel positive contributions of pH and oxygen and a negative contribution of phosphate. The "discharge component" pinpoints dilution due to positive loading for water mass and negative loadings for most element concentrations. Barborowski et al. [15,16] used multivariate statistical methods to assess the water quality at the monitoring station Magdeburg during low water and flood water conditions, respectively. They found seasonal phytoplankton development and the connected changes in redox conditions as well as tributaries to be the dominating factors during low water conditions, while resuspended contaminated sediments and reduced influence of the salt pollution of the Saale due to dilution were the dominating factors during flood conditions.

Despite these efforts and measures, research programs regarding the pollution of the Elbe River remain relevant, since they provide new insights for the conception of water management plans [17]. Current water monitoring concepts in Europe are based on the EU Water Framework Directive established in the year 2000. The goal of the directive is the protection and restoration of clean water in Europe and to ensure its long-term sustainable use [18]. Monitoring programs are the main tool to classify the status of each water body and serve as the basis for river management strategies [19]. In particular, highly resolved, reliable in situ water quality data are essential to guide priorities for investment and assess the need for protection and restoration of aquatic ecosystems to ensure the sustainable use of river systems in the future.

The intent of this study is to provide a comprehensive picture of the water dynamics and influences of tributaries in the measured stretch of the Elbe River and to identify and delineate sources influencing phytoplankton dynamics. In particular, the utilization of an in situ online multisensor system such as the BIOFISH device gives new insights by providing continuous, high temporal resolution monitoring data of rivers, which single point samples are unable to offer. Correlation analysis and the multivariate statistical methods cluster analysis (CA) and principal component analysis (PCA) are used to investigate sources of water quality variations and to assess the influence of phytoplankton dynamics, tributaries, and other pollution sources. Results are interpreted in the context of climate data acquired from the German Meteorological Service (DWD) and discharge data from the River Basin Community Elbe (FGG Elbe) to account for the complex dynamics of the river system.

## 2. Materials and Methods

### 2.1. The "Elbschwimmstaffel"

The joint science and sports event "Elbschwimmstaffel" in June/July 2017, which was conducted as part of the Year of Science 2016/17 under the slogan "Seas & Oceans", was conducted to raise public awareness and attention regarding the water quality of the Elbe River and rivers in general. In a unique survey of the Elbe River, in situ and online water quality data were acquired along a 550 km long stretch of the Elbe River from Dresden downstream to Geesthacht. The individual stages of the survey, which took place from 25 June 2017 to 12 July 2017, are illustrated in Figure 1. A long stretch of the planned river section was covered on 27 June 2017 to avoid problems caused by decreasing water levels. The period from 29 June 2017 to 2 July 2017 was spent in the port of Aken (Figure 1), because the research vessel had to wait for the swimmers of the sports event to catch up for the remaining survey.

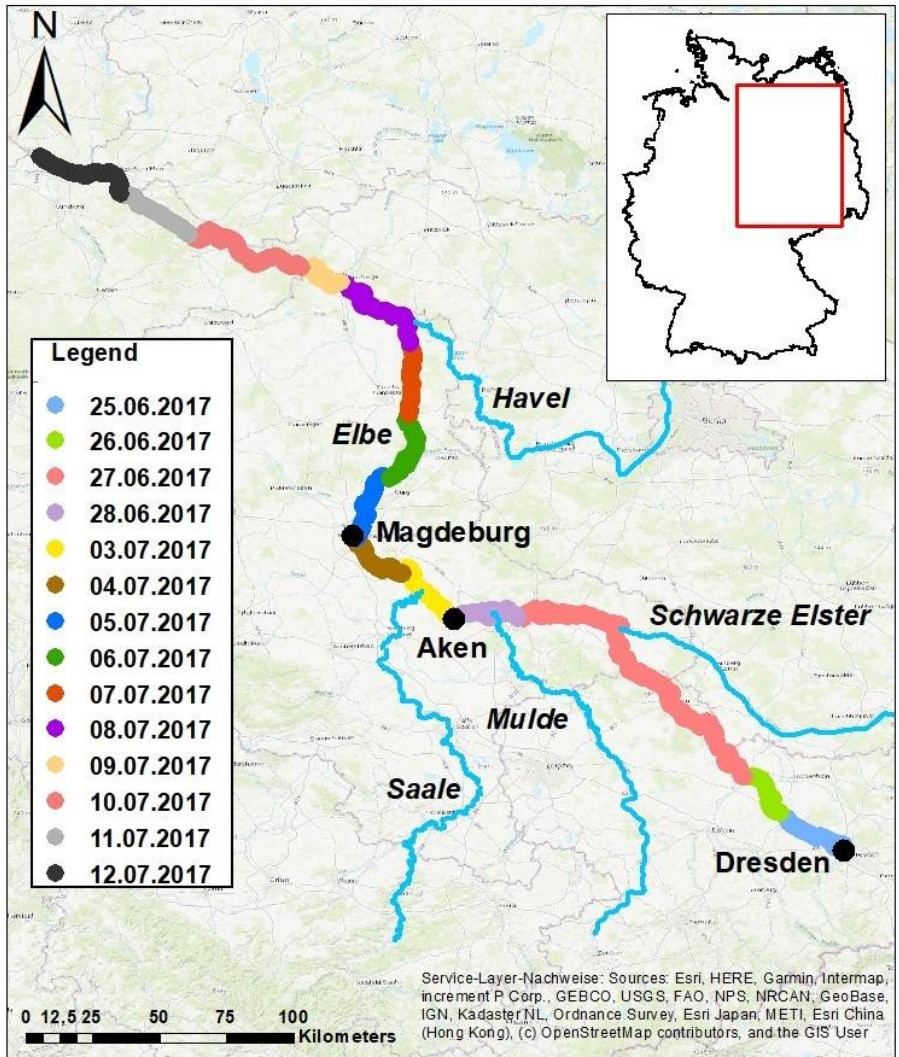

**Figure 1.** The investigated stretch of the Elbe River and major tributaries. Individual stages of the survey are colored according to date. The survey took place from 25 June 2017 to 12 July 2017. The survey covered a 550 km long stretch of the Elbe River from Dresden downstream to Geesthacht.

### 2.2. Study Area

#### 2.2.1. The Elbe River

The Elbe River has its source in the Giant Mountains in the Czech Republic before it runs through eastern and northern Germany and disembogues into the North Sea at

Cuxhaven after a length of 1094 km [10]. In Germany, the river has a length of 727 km [20]. The Elbe is the fourth largest river basin in Central Europe next to the Danube, the Vistula and the Rhine River. Almost 25 million people live in the Elbe river basin [21]. The main tributaries in Germany include the Schwarze Elster (Black Elster), the Mulde, the Saale and the Havel [22]. The investigated river stretch reaches from Dresden to Geesthacht as shown in Figure 1.

Weather data in Figure 2 depict precipitation, sunshine duration and mean air temperature during the period of the Elbschwimmstaffel. The climate data were retrieved from the Climate Data Center (CDC) of the German Weather Service [23]. The depicted parameters were selected from five DWD stations (Dresden-Klotzsche, Wittenberg, Magdeburg, Seehausen and Boizenburg) along the course of the Elbe, according to the position of the research vessel and its distance to the nearest station on the respective date. The daily mean temperatures varied between 14.6 °C and 24.0 °C. Relatively high temperatures were prevalent during the first two days of the Elbschwimmstaffel, as well as on 6 and 7 July 2017. Daily sunshine duration varied between 0.3 and 15.4 h and showed distinct variation between the different days. Periods of very low sunshine duration were recorded from 29 June 2017 to 2 July 2017 and from 10 July 2017 to 12 July 2017. Precipitation was limited to a few days. The highest precipitation took place on 10 July 2017, with 32.8 mm. Other dates with high precipitation were 28 July 2017 and 29 June 2017, with 16.4 mm and 13.9 mm, respectively.

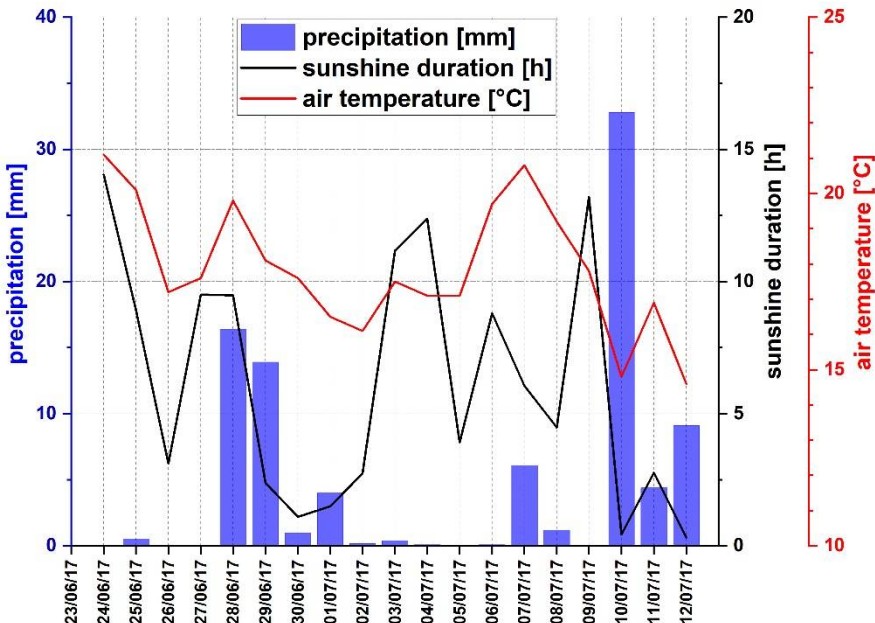

**Figure 2.** Precipitation [mm], sunshine duration (h) and mean air temperature at 2 m above ground (°C) during the measuring period. Data were extracted from the Climate Data Center (CDC) of the German Weather Service (DWD) [23]. The illustrated climate data are composed of climate data measured at the weather stations Dresden-Klotzsche, Wittenberg, Magdeburg, Seehausen and Boizenburg.

2.2.2. Discharge Data

Discharge data were retrieved from the Elbe Data Information System (FIS) of the River Basin Community Elbe (FGG Elbe) [24]. Figure 3 depicts the daily mean discharge rates of the Elbe at the monitoring stations Dresden, Magdeburg and Wittenberge during June and July 2017, normalized to the respective median of the daily mean discharge between 1997 and 2017. It shows below average discharge rates for the first few days of the Elbschwimmstaffel. At 30 June 2017, the monitoring station in Dresden showed a steep increase in discharge, which lasted until 2 July 2017. This discharge event propagated and reached the city of Magdeburg on 3 July 2017, overtaking the research vessel. When the

research vessel reached Magdeburg on 4 July 2017, the discharge rates were similar to the long-term median. The date of 30 June 2017 also marked the beginning of a separate discharge increase at Wittenberge. The research vessel reached Wittenberge at 8 July 2017, when the discharge rates were still close to their maximum.

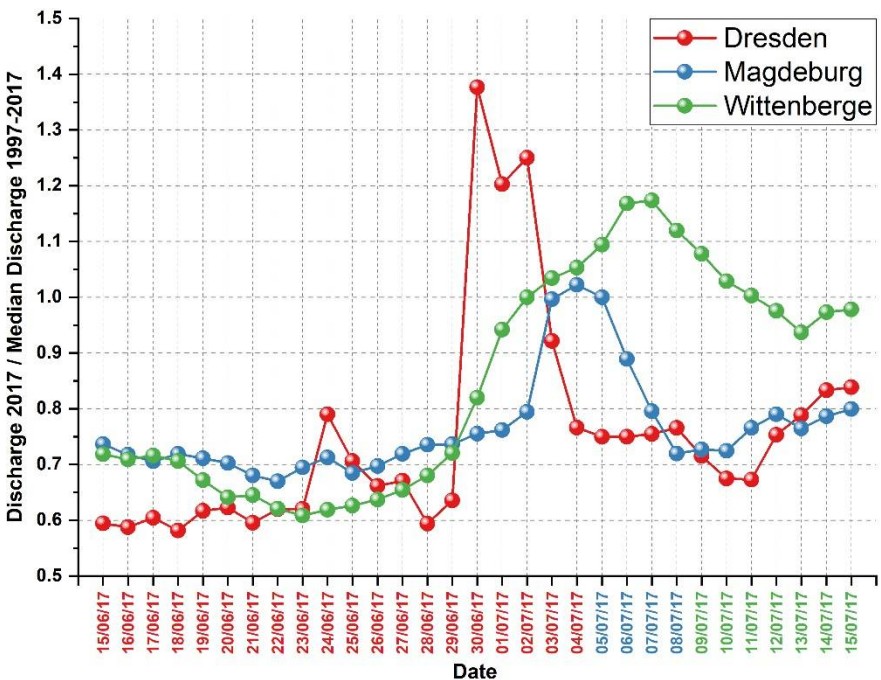

**Figure 3.** Normalized daily mean discharge at the monitoring stations Dresden, Magdeburg and Wittenberge from 15 June to 15 July 2017 (Discharge 2017). The respective median of the daily mean discharge between 1997 and 2017 (Median Discharge 1997–2017) was used to normalize the mean discharge rates of each day. The data were acquired from the Elbe Data Information System (FIS) [24] of the River Basin Community Elbe (FGG Elbe). The different colors of the date indicate the monitoring station that was most recently passed by the research vessel.

Figure 4 shows that the Mulde and Saale tributaries, which were passed on 26 June and 3 July, respectively (Figure 1), did not contribute significantly to the rising discharge rates of the Elbe. Daily mean discharge rates were for the most part below average. However, Havel shows, starting at 30 June 2017, discharge rates many times higher than the long-term median. This increase in discharge can be attributed to heavy rainfall in the Berlin region. A press release of the German Weather Service reports for the summer of 2017 the highest amount of precipitation ever recorded in the Berlin region (420 L/m$^2$). The extreme precipitation was mainly attributed to the low-pressure area RASMUND, which affected the Berlin area on 29 June 2017 [25].

*2.3. Physiochemical Parameters Measured by the BIOFISH*

The BIOFISH device (manufactured by ADM Elektronik, 23827 Krems II, Germany) is a multisensor system that can acquire in situ and online water quality data with high spatial and temporal resolution [26]. During the survey, eight physiochemical water quality parameters were collected with a frequency of 4 Hz. These parameters include Electrical Conductivity at 25 °C (EC25) [µs/cm], Temperature (Temp) [°C], pH-value, Oxygen Saturation (O$_2$-sat) [%], Turbidity (Turb) [FTU = Formazin Turbidity Unit], Colored Dissolved Organic Matter (CDOM) [ppbQS (Quinine Sulfate)], Chlorophyll a Fluorescence (Chl$_a$) [µg/L], Pressure [dBar] (for depth information [m]) and photosynthetic active radiation (PAR) [mmol/(s·m$^2$)] [27]. By fixing the BIOFISH on a floating body, measurements could be conducted in a fixed depth of around 0.5 m. One of the extension cranes of the WSV

research vessel "Elbegrund" was used to deploy the BIOFISH in front of the ship. With this setup, influences by the research vessel itself were minimized.

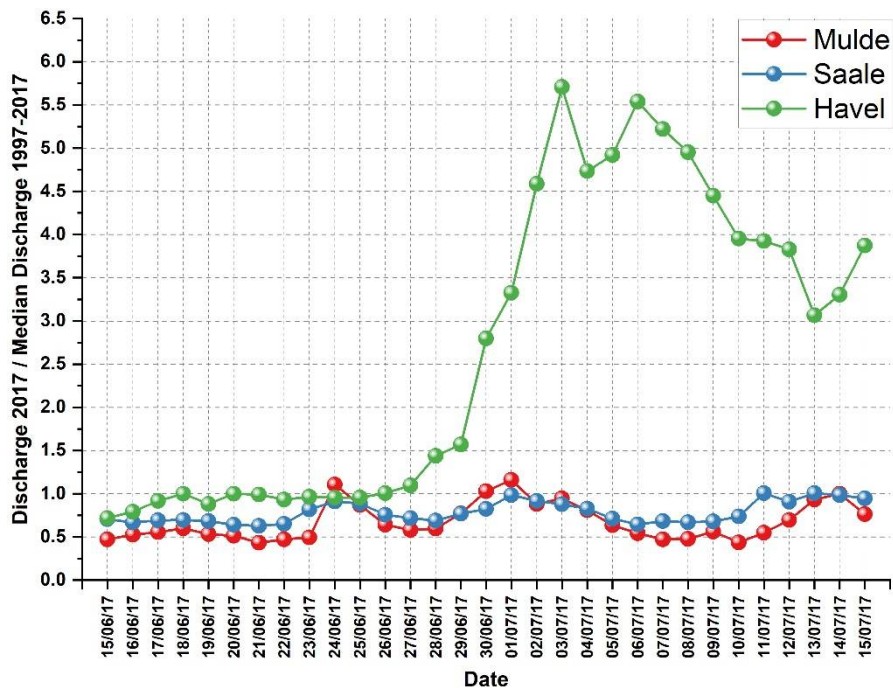

**Figure 4.** Normalized daily mean discharge of the tributaries Mulde (Bad Düben), Saale (Calbe-Griezehne) and Havel (Havelberg) from 15 June to 15 July 2017. The respective median of the daily mean discharge between 1997 and 2017 was used to normalize the mean discharge rates of each day. The data were acquired from the Elbe Data Information System (FIS) [24] of the River Basin Community Elbe (FGG Elbe).

For further evaluation, the data were binned into median values for each minute, as the median is less sensitive to outliers and skewed data than the mean value [28]. In general, simultaneous measurements of underwater *PAR* and above water *PAR* at varying depths can be used to derive the extent of the euphotic zone [29]. However, BIOFISH data of the Elbe River were recorded at a fixed depth of around 0.5 m. For further evaluation in this study, underwater *PAR*, above water *PAR* and depth were integrated into the new parameter Light Attenuation (*LA*). *LA* was calculated as shown in Equation (1):

$$\text{Light Attenuation } (LA) \left[\frac{1}{m}\right] = log10\left(\frac{\text{above water } PAR}{\text{underwater } PAR}\right)/\text{depth} \tag{1}$$

*LA* is based on the simple Lambert–Beer model, which is adequate for the estimation of light attenuation in shallow water bodies [30]. Due to the turbidity sensor only measuring the backscattering of light caused by suspended particulate matter, *LA* also integrates effects caused by CDOM and phytoplankton pigments.

### 2.4. Statistical Analysis

To interpret the BIOFISH data, a multivariate analysis including principal component and cluster analyses was conducted using SPSS statistics (v27). Prior to statistical analysis, z-standardization (mean = 0, sd = 1) of the dataset was performed to ensure that variables in different units and concentration ranges are comparable and to equalize their weights on analysis results.

Principal component analysis (PCA) can reduce large numbers of variables into a smaller number of new orthogonal, uncorrelated variables called principal components (PC), which will account for much of the variance in the original variables [31]. The purpose of PCA is

the reduction of dimensionality, which helps to find underlying impact factors and processes by simplifying complex and diverse relationships existing among observed parameters by revealing unobservable links between them [28,32,33]. If data have been measured in different physical units, differ by several orders of magnitude or the influence of elements with high variance is to be reduced, PCA is performed after standardization of variables. Due to the standardization of the variables, PCA was performed on the correlation matrix instead of the covariance matrix [34]. Further, Kaiser's Measure of Sampling Adequacy (MSA) was conducted to examine the suitability of the datasets for PCA.

Cluster analysis (CA) was used to find patterns in the dataset by grouping observations into a finite number of clusters [35]. It results in similar observations in each cluster while the clusters themselves are dissimilar to each other [36]. Hierarchical cluster analysis was performed on the normalized datasets using the cluster Ward's method and the distance type Euclidean. Additionally, k-means clustering was performed on the observations of each dataset. The results of the hierarchical cluster analysis were used to specify the number of clusters and the initial cluster centers used by the k-means algorithm in advance. The output consists of the cluster membership of each observation and the final cluster centers.

### 2.5. Quality Assurance of BIOFISH Data

The 5th to 95th percentile ranges of every minute of recorded BIOFISH data were evaluated to check how well each parameter is represented by the mean of every minute. Descriptive statistics of the 5th to 95th percentile ranges are listed in Table 1. The parameter PAR has a relatively high mean variation, probably due to rapid and strong changes in incoming radiation caused by clouds or bridges within a period of one minute. Influences on water parameters such as tributaries are probably responsible for observations with very large 5th to 95th percentile ranges. The overall relatively small mean 5th to 95th percentile ranges indicate that the median of every minute is suited to represent most of the BIOFISH data.

**Table 1.** Mean, standard deviation (SD), minimum and maximum of the 5th to 95th percentile ranges of every minute of BIOFISH data (n = 2152).

| Parameter | Pressure | Temp | $EC_{25}$ | $O_2$ | pH | $Chl_a$ | CDOM | Turb. | $PAR_{uw}$ | $PAR_{aw}$ |
|---|---|---|---|---|---|---|---|---|---|---|
| Unit | dbar | °C | mS/cm | % | - | µg/L | ppb | FTU | $mmol/(s·m^2)$ | $mmol/(s·m^2)$ |
| Mean | 0.12 | 0.03 | 0.01 | 1.03 | 0.02 | 4.64 | 0.86 | 0.18 | 168 | 91.4 |
| SD | 0.04 | 0.08 | 0.04 | 2.03 | 0.05 | 4.42 | 2.83 | 0.23 | 321 | 126 |
| Minimum | 0.05 | 0.00 | 0.00 | 0.11 | 0.00 | 0.51 | 0.22 | 0.04 | 0.00 | 0.51 |
| Maximum | 0.34 | 2.07 | 0.77 | 37.0 | 1.34 | 106 | 101 | 7.19 | 1620 | 953 |

## 3. Results

### 3.1. BIOFISH Data

#### 3.1.1. Longitudinal Profiles

Descriptive statistics of the BIOFISH data are listed in Table 2.

**Table 2.** Descriptive statistics of the BIOFISH data (n = 2152).

| Parameter | Unit | Mean | SD | Minimum | Median | Maximum |
|---|---|---|---|---|---|---|
| Temp | (°C) | 22.3 | 0.9 | 20.4 | 22.4 | 24.2 |
| $EC_{25}$ | (mS/cm) | 0.80 | 0.28 | 0.45 | 0.91 | 1.73 |
| $O_2$% | (%-sat.) | 137.3 | 28.3 | 92.3 | 130.8 | 277.2 |
| pH | - | 8.9 | 0.3 | 7.6 | 8.9 | 9.5 |
| $Chl_a$ | (µg/L) | 68.0 | 22.5 | 7.5 | 64.3 | 119.8 |
| CDOM | (µg/L) | 45.2 | 8.7 | 36.6 | 42.9 | 141.1 |
| Turb | (FTU) | 3.1 | 0.8 | 0.8 | 3.3 | 4.9 |
| LA | - | 2.0 | 0.5 | 0.6 | 1.9 | 4.1 |

Figure 5 illustrates the changes in $EC_{25}$, $Chl_a$, Turb and LA along the course of the investigated stretch of the Elbe River. Significant changes during the period of the survey can be observed for all parameters recorded by the BIOFISH system (Table 2). Sudden changes coincide often with the inflow of tributaries, especially the Saale (EC25) and the Havel (EC25, CDOM, Turb, LA).

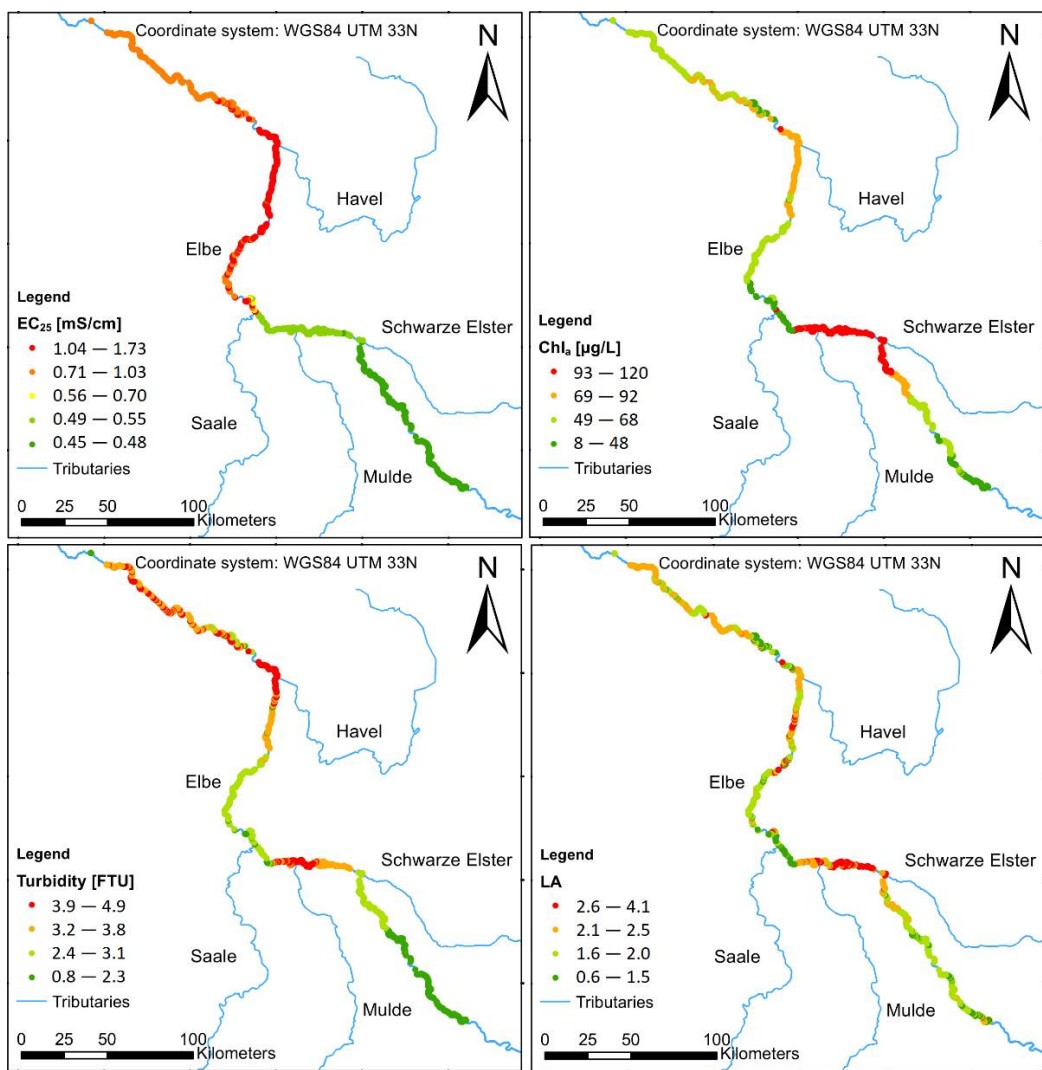

**Figure 5.** $EC_{25}$ (**top left**), $Chl_a$ (**top right**), Turbidity (**bottom left**) and LA (**bottom right**) concentrations along the investigated stretch of the Elbe River. The class ranges are based on the "Natural Breaks" method of ArcGIS. The colors used for classification are not related to the water quality.

### 3.1.2. Correlation Analysis

Table 3 shows the correlation matrix of the BIOFISH parameters. Due to the high number of degrees of freedom (df = 2150), a very low Pearson's r is considered statistically significant (r value of about 0.42 at $p$-value = 0.05) [33]. Hence, instead of the usual approach to search for statistically significant correlations ($p$-value < 0.05), only correlations with a Pearson's r greater than 0.5 are considered for further evaluation.

**Table 3.** Correlation matrix of eight physicochemical BIOFISH parameters. Underlined values represent that correlation is significant at the 0.05 level. Due to the high number of samples (n = 2152), even correlations with very low Pearson's r have *p*-values below 0.05. Correlations with a Pearson's r greater than 0.5 are therefore bold.

|                | Temp  | $EC_{25}$ | $O_2\%$ | pH    | $Chl_a$ | CDOM  | Turb | LA   |
|----------------|-------|-----------|---------|-------|---------|-------|------|------|
| Temp           | 1.00  |           |         |       |         |       |      |      |
| $EC_{25}$      | −0.36 | 1.00      |         |       |         |       |      |      |
| $O_2\%$        | 0.37  | −0.37     | 1.00    |       |         |       |      |      |
| pH             | 0.12  | −0.13     | **0.74**| 1.00  |         |       |      |      |
| $Chl_a$        | 0.20  | −0.11     | **0.56**| **0.86**| 1.00  |       |      |      |
| CDOM           | −0.20 | 0.17      | **−0.50**| **−0.67**| −0.47 | 1.00 |      |      |
| Turb           | −0.04 | 0.48      | 0.08    | 0.45  | **0.69**| −0.13 | 1.00 |      |
| LA             | 0.03  | −0.02     | 0.35    | 0.46  | **0.59**| −0.13 | 0.47 | 1.00 |

High positive correlations can be observed for $Chl_a$ and $O_2\%$ (r = 0.56), pH (r = 0.86), Turb (r = 0.69), and LA (r = 0.59). A high positive correlation can also be observed for pH and $O_2\%$ (r = 0.74). CDOM shows negative correlations with $O_2\%$ (r = −0.50) and pH (r = −0.67). No correlations with a Pearson's r greater than 0.5 can be observed for the parameters Temp and $EC_{25}$.

The correlations between $Chl_a$ and the parameters $O_2\%$, pH and Turb are illustrated as scatter plots in Figure 6. Coloration by sampling date shows that the nearly linear increase of $Chl_a$ together with $O_2\%$, pH and Turb has different gradients on different sampling dates.

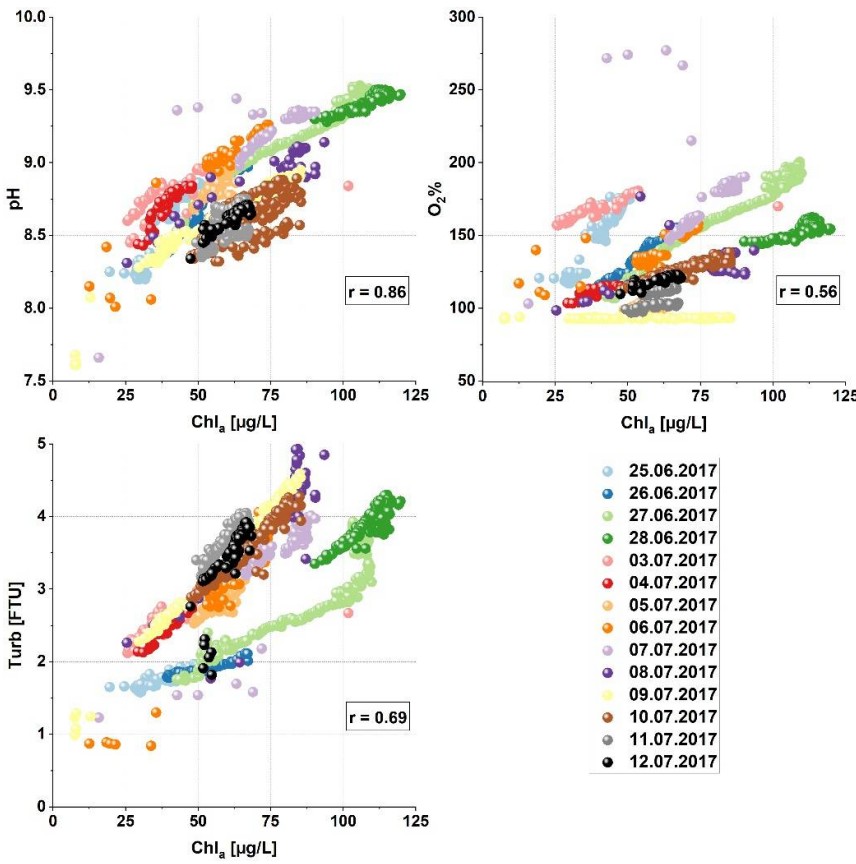

**Figure 6.** Scatter plots representing the correlation between Chlorophyll a ($Chl_a$) and pH (r = 0.86), $O_2\%$ (r = 0.56) and Turbidity (Turb) (r = 0.69). Observation points are colored according to the sampling date. Varying linear relationships between $Chl_a$ and the other BIOFISH parameters can be observed for the different sampling dates.

3.1.3. Cluster Analysis (CA)

The parameters Temp and $EC_{25}$ were not included in the CA due to the results of the correlation analysis showing no significant correlations with other parameters (Table 3). CA was applied to identify similarities within the BIOFISH dataset. Three clusters were determined based on the results of the hierarchical cluster analysis (Dendrogram for BIOFISH observations (n = 2152) in Supplementary Material Figure S1), which allowed the classification of the dataset into several distinct river sections. The observation points prior to the mouth of the Havel belong almost exclusively to cluster 1 and 3 (Figure 7). In contrast, most of the observation points downstream of the mouth of the Havel were assigned to cluster 2. It is noticeable that the spatial distribution of cluster 1 and cluster 3 is not associated with river mouths of major tributaries.

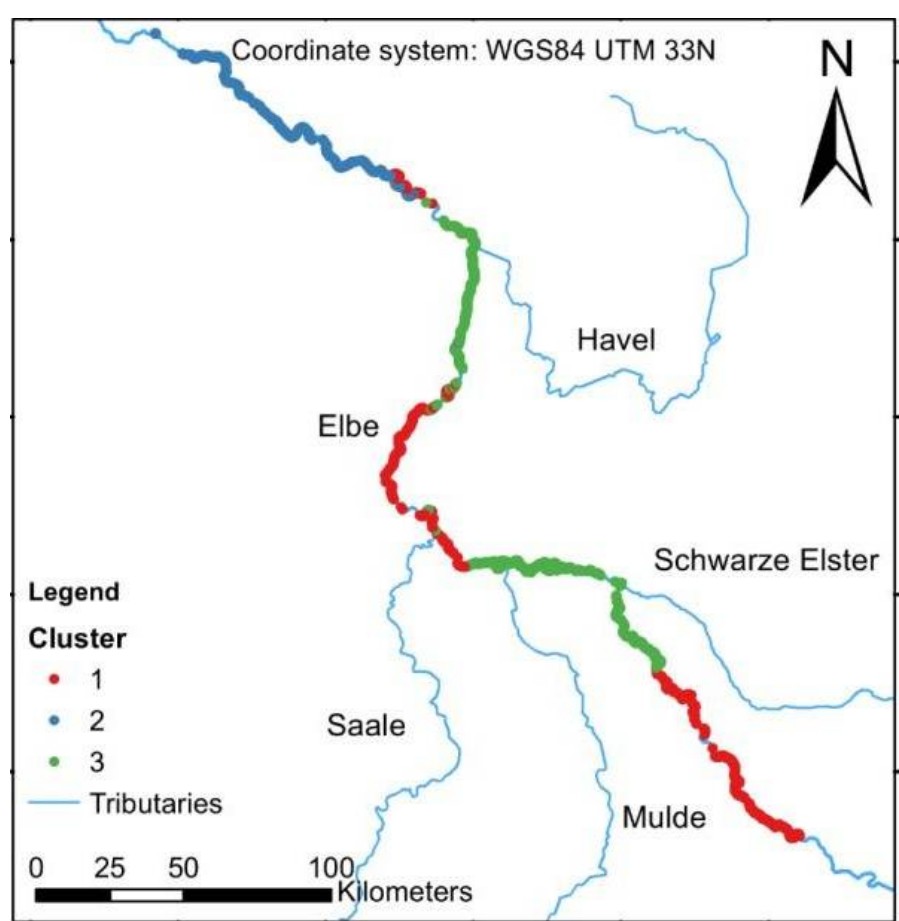

**Figure 7.** Spatial cluster membership distribution obtained from k-means clustering performed on BIOFISH parameters. Observation points are colored according to cluster membership.

Figure 8 depicts the standardized cluster centers of each cluster after k-means clustering. Cluster 1 shows below average values for most of the BIOFISH parameters. In particular, $Chl_a$, Turb and LA values are very low compared to the other cluster. While the $O_2$% and pH values of cluster 2 are even lower than in cluster 1, Turb and LA values are relatively high and the CDOM value is much higher than in the other clusters. Cluster 3 stands out with very high values of the parameters $O_2$%, pH, $Chl_a$, Turb and LA. Only the CDOM value is low in cluster 3.

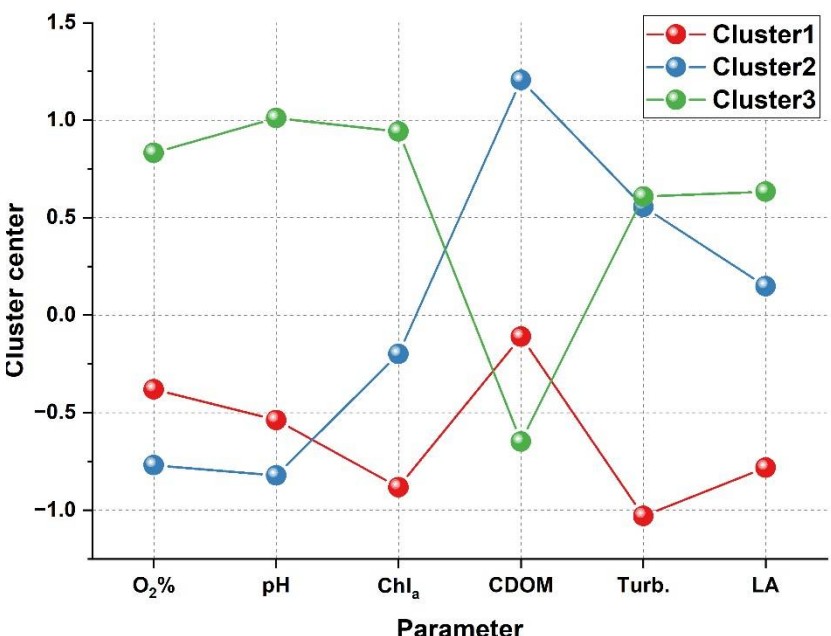

**Figure 8.** Plot of standardized cluster centers obtained from k-means clustering performed on BIOFISH parameters. A value of 0 corresponds to mean concentration and a value of 1 to standard deviation. The cluster centers indicate characteristic values of each cluster.

### 3.1.4. Principal Component Analysis (PCA)

The suitability of the BIOFISH dataset for PCA was tested by Kaiser's measure of sampling adequacy (MSA). The parameters $EC_{25}$ and Temp had to be removed from the BIOFISH dataset, since their low MSA values (<0.5) indicated that they were not suitable for PCA. Overall MSA of the remaining parameters (0.73) proved that PCA can achieve significant reduction of the dimensionality of the BIOFISH dataset. Two PCs were retained based on scree plot (Supplementary Material Figure S2) and Kaiser's criterion (eigenvalues > 1). They explain 78.52% of the variance contained in the original dataset.

Table 4 lists the loadings of the BIOFISH parameters on the varimax-rotated PCs. PC 1, responsible for 41.06% of the total variance, has strong positive loadings on $O_2$% (0.85) and pH (0.83), a moderate positive loading on $Chl_a$ (0.56) and a strong negative loading on CDOM (−0.85). PC 2 accounts for 37.46% of the total variance, with strong positive loadings of $Chl_a$ (0.77), Turb (0.90) and LA (0.77). In Figure 9, the relationship between PC scores and cluster memberships of the BIOFISH observations is illustrated. The plot shows a clear distinction between the three clusters. Observations belonging to cluster 1 show moderate PC 1 scores and PC 2 scores vary between moderate and very low. Cluster 2 observations show lower PC 1 scores and higher PC 2 scores. The observations of cluster 3 show generally higher PC 1 and PC 2 scores.

**Table 4.** Loadings of BIOFISH parameters on varimax-rotated PCs. Moderate loadings (0.5–0.75) are bold; strong loadings (>0.75) are bold and underlined.

| Variable | PC 1 | PC 2 |
|:---:|:---:|:---:|
| $O_2$% | **<u>0.85</u>** | 0.16 |
| pH | **<u>0.83</u>** | 0.49 |
| $Chl_a$ | **0.56** | **<u>0.77</u>** |
| CDOM | **<u>−0.85</u>** | −0.02 |
| Turb | 0.01 | **<u>0.90</u>** |
| LA | 0.18 | **<u>0.77</u>** |
| Eigenvalue | 2.46 | 2.25 |
| % Variance explained | 41.06 | 37.46 |
| % Cumulative Variance | 41.06 | 78.52 |

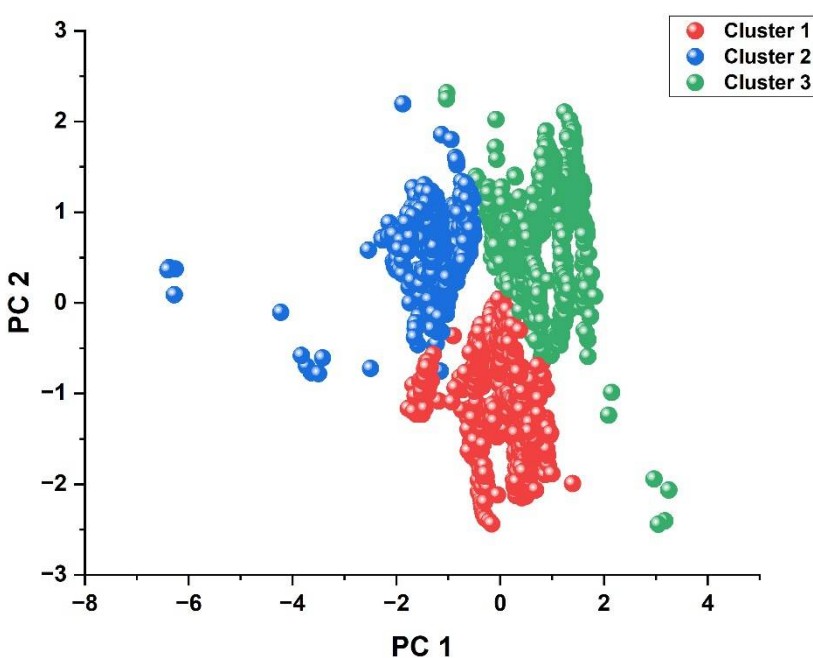

**Figure 9.** Scatter plot of BIOFISH PC scores after varimax rotation. PC 2 scores are plotted against PC 1 scores. Observations are colored according to the CA results. The observations of the three clusters are clearly differentiated by their respective PC scores.

Figure 10 illustrates the change of PC scores along the Elbe River. PC 1 scores show an approximately linear increase over the course of the first three survey days (until river km 107). This correlation between PC 1 and time of the day is statistically significant (*p*-value < 0.05). PC 2 scores show only a weak increase over the course of the first three days and are lower than the PC 1 scores. In contrast to PC 1, PC 2 scores are not significantly correlated to time of the day. PC 2 scores show a steep increase after river km 150 on the fourth day of the survey. At river km 200, which is shortly after the mouth of the Schwarze Elster tributary, PC 1 scores start to decrease while PC 2 scores are fluctuating. After reaching the port of Aken (river km 277) on the fifth day of the survey and the following longer stop, PC 1 scores and PC 2 scores again are much lower. They increase again over the course of the following days. River km 438, where the Havel flows into the Elbe River, marks a significant change in the scores of both PCs. PC 2 scores show high fluctuation but remain relatively high, while PC 1 scores remain relatively low.

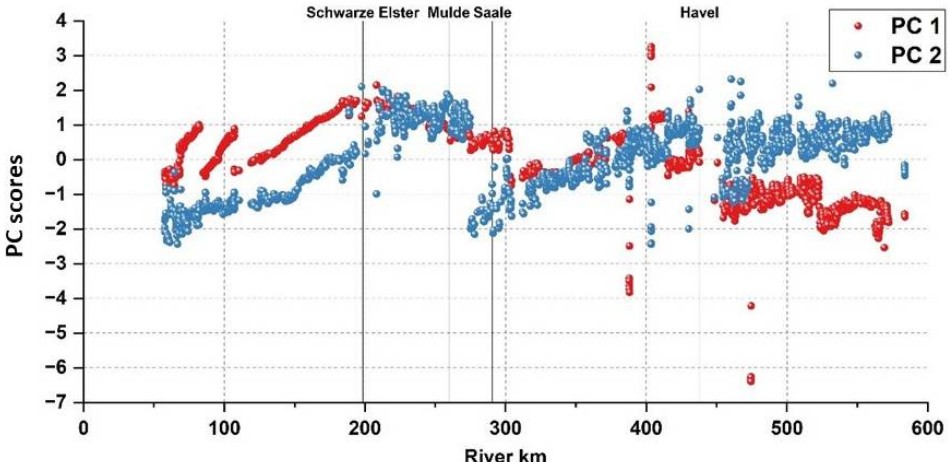

**Figure 10.** BIOFISH PC scores after varimax rotation plotted against river km of the Elbe River.

### 3.2. Evolution of $EC_{25}$ over the Course of the Survey

Noticeable changes in $EC_{25}$ were identified by plotting the $EC_{25}$ values measured by the BIOFISH against river km of the Elbe. Figure 11 illustrates changes in $EC_{25}$ along the investigated stretch of the Elbe River. Prior to the mouth of the Saale, $EC_{25}$ values remain relatively constant at a level of around 0.5 mS/cm. At the mouth of the Saale, $EC_{25}$ increases sharply and reaches, with >1.5 mS/cm, over three times higher values than before. In the following course, after the mouth of the Saale, BIOFISH observations show significant variation, which decreases steadily with increasing distance to the Saale tributary.

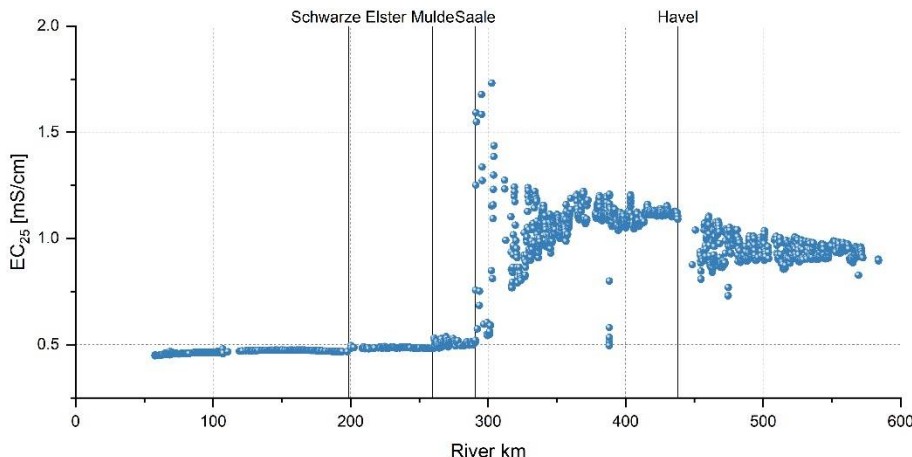

**Figure 11.** $EC_{25}$ plotted against river km of the Elbe River. Major tributaries are marked with black lines. A significant increase and high variation in $EC_{25}$ can be observed after the mouth of the Saale.

A distinct decrease in $EC_{25}$ along with higher variation of the recorded values can also be observed after the mouth of the Havel. By plotting $EC_{25}$ prior to the mouth of the Saale and after the mouth of the Saale separately, small changes and characteristics in $EC_{25}$ are easier to identify.

Figure 12 shows the first half of the surveyed Elbe section between Dresden and the mouth of the Saale. $EC_{25}$ values prior to the mouth of the Schwarze Elster vary between 0.45 mS/cm and 0.48 mS/cm. A small increase in $EC_{25}$ values can be observed after the research vessel passed the wastewater treatment plants at Dresden Kaditz and Nünchritz (river km 64 and 103, respectively). An increase in $EC_{25}$ can also be observed at river km 85 and 119, where the research vessel stopped during the night on the first days of the measurement campaign. In the morning, when the recording of water quality parameters resumed, measured $EC_{25}$ values were higher than on the previous day, resulting in this noticeable offset. At river km 199, where the Schwarze Elster flows into the Elbe River, $EC_{25}$ values increase from about 0.46 mS/cm to values between 0.48 mS/cm and 0.49 mS/cm. Furthermore, $EC_{25}$ values show an additional significant increase after the mouth of the Mulde and high variation between 0.48 mS/cm and 0.53 mS/cm. This variation decreases towards the mouth of the Saale but stays relatively high over a length of about 30 km.

The second half of the surveyed Elbe section between the mouth of the Saale and Geesthacht is shown in Figure 13. As mentioned above, $EC_{25}$ values rise very sharply from around 0.51 mS/cm before the river mouth of the Saale up to 1.75 mS/cm afterwards, and start to fluctuate strongly between 0.54 mS/cm and 1.74 mS/cm. Although a steady decrease in this variation can be observed thereafter, even 160 km after the mouth of the Saale, at river km 350, this impact remains noticeable and $EC_{25}$ values are still varying between approximately 1.00 mS/cm and 1.20 mS/cm. A general increase to $EC_{25}$ values between 1.08 mS/cm and 1.23 mS/cm can further be observed after river km 356. This increase coincides with the start of the BIOFISH recordings on 6 July 2017. However, this location also falls together with a major spoil heap near Zielitz, where the K + S chemical company dumps byproducts of potash mining. Observation points at river km 388, where

EC$_{25}$ values are comparatively low, were recorded in the harbor of Tangermünde, where the tributary Tanger flows into the Elbe River. A location with slightly higher EC$_{25}$ values was also recorded at river km 403 in the harbor of Arneburg. Prior to the mouth of the Havel, EC$_{25}$ values are relatively constant (around 1.15 mS/cm). EC$_{25}$ values recorded after the mouth of the Havel tend to be lower, but vary significantly more than before, between 0.80 mS/cm and 1.10 mS/cm. As for the other tributaries, this fluctuation decreases with increasing distance from the mouth of the Havel. With the exception of this, no noticeable changes in EC$_{25}$ can be observed for this last stretch of the investigated part of the Elbe River. Close to Geesthacht, the EC$_{25}$ values recorded at this stage of the survey vary between 0.88 mS/cm and 0.98 mS/cm.

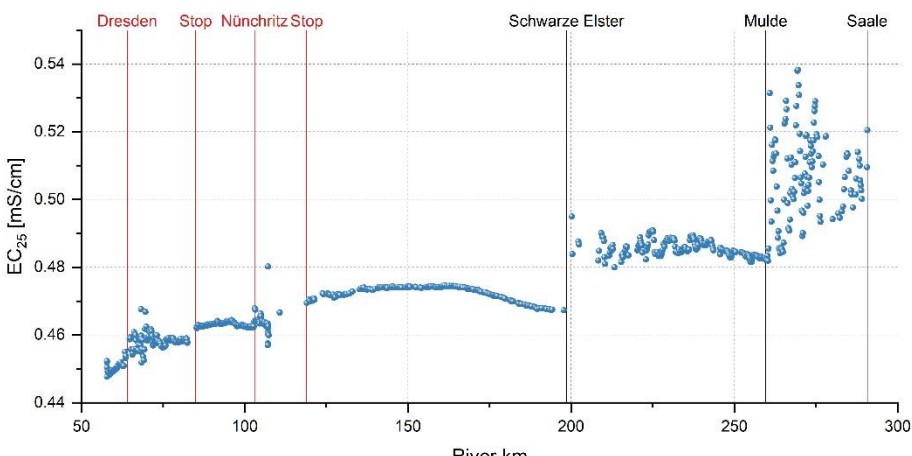

**Figure 12.** EC$_{25}$ plotted against river km of the Elbe. Only the investigated stretch of the Elbe River prior to the mouth of the Saale is depicted. Major tributaries are marked with black lines. An increase in EC$_{25}$ values can be observed after the mouth of the Schwarze Elster and the Mulde. Red lines mark noticeable changes in EC$_{25}$ unrelated to tributaries.

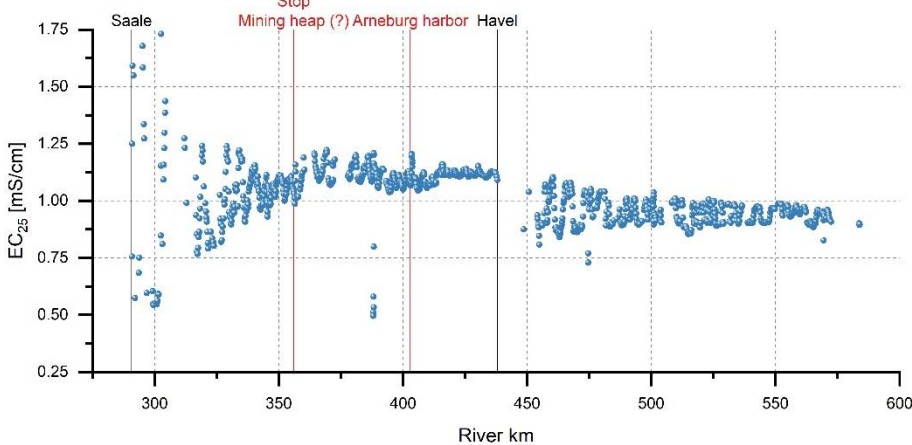

**Figure 13.** EC$_{25}$ plotted against river km of the Elbe. Only the investigated stretch of the Elbe River after the mouth of the Saale is depicted. Major tributaries are marked with black lines. Red lines mark a noticeable increase in EC$_{25}$ unrelated to tributaries. The recorded EC$_{25}$ values show generally high variations in this part of the Elbe River.

## 4. Discussion

### 4.1. Phytoplankton Dynamics of the Elbe River

The parameters recorded by the BIOFISH system allow for a comprehensive analysis of phytoplankton dynamics in rivers. Chl$_a$ can be used to estimate phytoplankton biomass [37], while primary production and respiration are related to changes in O$_2$% and pH, and Turb and LA reflect the light conditions in rivers [38].

High correlations were determined between $Chl_a$, $O_2$% and pH, with high values reflecting increasing photosynthetic activity with growing phytoplankton biomass [39]. The differently colored sampling dates in Figure 6 illustrate distinct variations in $Chl_a$, $O_2$% and pH values due to the diurnal cycle, with relatively low values in the morning and much higher values in the evening. Days with high photosynthetic activity resulted in relatively high maximum $O_2$% (277%) and pH (9.5) values. The high correlations observed between $Chl_a$, Turb and LA suggest that phytoplankton cells worsen light penetration and contribute significantly to the turbidity in the Elbe River. The spatial influence of these correlations is also significantly visible over many kilometers and several days in different segments during the survey (Figure 5). Temp shows only relatively weak correlations with the other BIOFISH parameters (Table 3). While a relationship between phytoplankton levels and water temperature was previously reported [21,40], low correlation between Temp and $Chl_a$ can probably be attributed to different response times of Temp and $Chl_a$ to changes in global radiation as well as to the rather low fluctuation of the absolute water temperature in the measurement period (Table 2). The negative correlation that can be observed between CDOM and the parameters $Chl_a$, $O_2$% and pH is probably related to the degradation of phytoplankton as a source of CDOM [41].

Sections of the Elbe River with different levels of phytoplankton biomass and photosynthetic activity were identified with CA (Figure 7). The normalized cluster centers show that observations associated with cluster 1 represent sections of the Elbe River with low levels of phytoplankton biomass (low $Chl_a$, Turb and LA values) and photosynthetic activity (low $O_2$% and pH values), while cluster 3 observations represent sections with high levels of phytoplankton biomass and photosynthetic activity (Figure 8). Cluster 2 observations, which make up most of the section after the mouth of the Havel, are characterized by low levels of photosynthetic activity, moderate levels of phytoplankton biomass, and high levels of CDOM, Turb and LA (Figure 8).

The results of the CA are supported by the results of the PCA, since the observations of each cluster can be discerned by a combination of their PC scores (Figure 9). PC 1, weighted positively on pH, $O_2$%, $Chl_a$, and negatively on CDOM, represents "photosynthetic activity", while PC 2, weighted positively on Turb, LA and $Chl_a$, represents "turbidity" and also "phytoplankton biomass".

Based on the cluster memberships of the observations (Figure 7) supported by PC score changes along the course of the Elbe River (Figure 10), a comprehensive study of the phytoplankton dynamics during the survey is possible. During the first three days of the survey, from the 25th of June 2017 to the 27th of June 2017, the membership in cluster 1 with low $Chl_a$ and Turb concentrations shows an overall low degree of photosynthetic activity and phytoplankton biomass. However, low discharge and turbidity in the Elbe River (Figure 3) coupled with favorable climate conditions (Figure 2) enabled higher photosynthetic activity in the diurnal cycle, also shown by increasing PC 1 scores over the course of the day (Figure 10). Due to the high distance covered on the 27 June 2017, a different waterbody than on the previous days was sampled in the afternoon of the 27th and on the 28th of June 2017. The differences in hydrological and meteorological history of this waterbody caused a change in cluster membership from cluster 1 to cluster 3, also characterized by increased PC 2 scores (Figure 10). The sampled waterbody coincides with the discharge peak measured on the 24th of June at Dresden monitoring station (Figure 3), when an average flow velocity of 2.34 km/h is considered (flow velocity for Dresden monitoring station at 150 $m^3$/s) [42]. The higher discharge measured on the 24 June at Dresden monitoring station (Figure 3) explains higher levels of turbidity and also phytoplankton biomass (PC 2) which in turn leads to a slight inhibition of photosynthesis (PC 1) (Figure 9) [43], even though the overall photosynthetic activity is still relatively high due to the higher amount of phytoplankton biomass.

A different water body was sampled once again on the 3rd of July 2017, when the research vessel resumed sampling after the stop in the harbor of Aken (Figure 1). Low PC 2 scores and association with cluster 1 indicate relatively low photosynthetic activity and

phytoplankton biomass (Figure 10). A likely cause is the unfavorable weather conditions with very low sunshine duration during the stop at Aken between the 29 June and the 2 July 2017 (Figure 2). From the 6th of July through the 8th of July 2017, a steady increase in PC 1 and PC 2, associated with cluster 3, can be observed, indicating favorable conditions for photosynthesis and growing phytoplankton biomass (Figures 9 and 10). The meteorological conditions during this period support this assumption. Sunshine duration was relatively high and only small precipitation events occurred (Figure 2). The highly increased discharge of the Havel due to heavy precipitation events (Figure 4) characterized the phytoplankton dynamics from the 9 July 2017 onward until the last stage of the survey on the 12 July 2017. Both cluster membership of the observations and PC scores recorded downstream of the mouth of the Havel point to unfavorable conditions for photosynthesis (Figures 7 and 10). High turbidity and CDOM values caused by heavy precipitation events (Figure 4), as well as very low sunshine duration, resulted in poor light conditions. As a result, $O_2$% and pH values stayed very low even though $Chl_a$ concentrations remained on a moderate level.

The influence of the tributaries Schwarze Elster, Mulde and Saale on phytoplankton dynamics is hard to assess due to the multitude of other factors that must be considered. No changes in cluster membership can be observed at the mouths of these tributaries (Figure 7) and noticeable changes in PC score occur only after the mouth of the Schwarze Elster (Figure 10). The negligible influence of these tributaries during the survey period coincides with the findings of Karrasch et al. [44], who found that no distinct differences in biological parameters were caused by the inflow of Mulde and Saale. Low $Chl_a$ concentrations in Mulde and Saale were reported by FGG Elbe [10] and attributed to the unfavorable light conditions along the courses of both tributaries. The constant increase of $Chl_a$ concentrations along the course of the Elbe River stated by Guhr et al. (2003) [21] and Eidner et al. [45] can be observed for sections of the survey, where hydrological and meteorological conditions remained relatively unchanged. $Chl_a$ concentrations increased from 60 to 110 µg/L over the course of the first three days of the survey and from 25 to 93 µg/L between the 6–8 July 2017. However, overall $Chl_a$ concentrations during the survey showed great variation due to changing hydrological and meteorological conditions.

### 4.2. Longitudinal River Surveys with the BIOFISH System

Previous longitudinal surveys of the Elbe River were carried out to assess changes in water quality [9,10,20], collect data for modeling purposes [46], or assess the effects of nutrient concentrations in the Elbe River [21]. In the course of these surveys, samples were taken at a limited number of measuring sites. The major difference to previous surveys is therefore the high spatial and temporal resolution of the parameters recorded by the BIOFISH system. The high resolution allowed for a detailed evaluation of BIOFISH parameters regarding spatial and temporal variations, instead of just observing general changes along the longitudinal profile of the Elbe River [20].

For example, diurnal variations in primary production and the effect on pH and $O_2$% were clearly visible (Figure 6). The high spatial resolution of $EC_{25}$ proved useful for identifying point-sources of pollution, although the unique characteristics of the Elbe River limited this approach (Figures 11–13). While wastewater treatment plants in the early stages of the survey were still partially identifiable by increased $EC_{25}$ values (Figure 11), $EC_{25}$ values in the middle part of the investigated stretch of the Elbe River showed high variation due to the salt pollution of the Saale water, which required more than 100 km to be fully mixed with the Elbe River water (Figures 12 and 13). Since the research vessel was accompanying the daily stages of the Elbschwimmstaffel, travel time was not adjusted to the flow time of the water. The sampling of different water bodies can cause misinterpretation of $Chl_a$ concentrations, since phytoplankton biomass is closely related to the meteorological and hydrological history of a water body [20].

The evaluation of the phytoplankton dynamics in the Elbe River confirmed that different $Chl_a$ concentrations in different water bodies had to be considered for interpretation. Noticeable changes of $EC_{25}$ were also related to different water bodies (Figure 11). The

interpretability of data collected by future longitudinal BIOFISH surveys could therefore be improved by sampling the same water body over the course of the survey (Lagrange approach) [21]. Previous longitudinal surveys of the Elbe River using Lagrangian sampling based their sampling strategy on flow times calculated with tracer experiments [21] or flow time models [46]. An international coordinated monitoring program, based on the guidelines of the Water Framework Directive of the EU, ensures extensive analysis of the water quality of the Elbe River and its tributaries [47].

While the extent of new insights gained by the BIOFISH survey in this regard was therefore limited by the event schedule, results show that longitudinal surveys using the BIOFISH system can be used to identify diverse problems in aquatic ecosystems such as algal blooms and sources of pollution. Accordingly, longitudinal BIOFISH surveys could be useful for rivers where the coverage of water quality data is poor. Further, high-resolution multisensor data from the BIOFISH system can also deliver significant contributions for modeling $Chl_a$ concentrations in rivers [48]. Insights gained by such surveys could serve as a basis for establishing effective water quality monitoring networks.

## 5. Conclusions

Water quality data, obtained along a 550 km stretch of the Elbe River from Dresden downstream to Geesthacht, were evaluated with the goal of gaining a comprehensive picture of the water quality of the Elbe River. The acquired datasets comprised eight water quality parameters recorded in high spatial and temporal resolution by the BIOFISH multisensor system. Multivariate statistical methods such as cluster analysis (CA) and principal component analysis (PCA) were used to reduce the dimensionality of these datasets and identify sources of variation in water quality. The parameters recorded by BIOFISH enabled a comprehensive study of phytoplankton dynamics over the course of the survey. Strong and significant ($p$-value $< 0.05$) Pearson correlations of $Chl_a$ with Turb ($r = 0.69$), $O_2\%$ ($r = 0.56$) and pH ($r = 0.86$) show that phytoplankton development and photosynthetic activity have major influence on the water quality of the Elbe River. Evaluation of the statistical analysis results in the context of hydrological data and weather data revealed that discharge and the meteorological conditions during the previous days were major influences on phytoplankton development. A strong increase of discharge after the inflow of the Havel, related to heavy precipitation events in the Berlin area, characterized phytoplankton dynamics during the last third of the survey. The results suggest that worsened light conditions in this section of the Elbe River due to high Turb and CDOM concentrations inhibited photosynthetic activity. A notable influence of Mulde and Saale on phytoplankton development in the Elbe River was not discernable. $EC_{25}$ values recorded by BIOFISH suggest that the Saale water required more than 100 km to be fully mixed with the Elbe water. Future longitudinal river surveys with the BIOFISH system could be adapted to the flow time of the river water for easier interpretation of the recorded parameters. In the case of this survey, good coverage of monitoring and weather stations along the Elbe River compensated at least partially for the impairment of the interpretability of the survey data by these factors. The general agreement of the gained results with those of other studies and reports shows that key processes impacting the water quality of the Elbe River were correctly identified. The high resolution of the parameters recorded by the BIOFISH system turned out to be particularly useful for characterizing phytoplankton dynamics and mixing processes as well as identifying sources of pollution. Longitudinal river surveys using the BIOFISH system, coupled with a sensible sampling strategy, should prove very valuable for assessing the water quality of rivers.

**Supplementary Materials:** The following supporting information can be downloaded at https://www.mdpi.com/article/10.3390/w14132078/s1, Figure S1: Dendrogram of the BIOFISH observations (n = 2152). Based on the result of hierarchical cluster analysis performed on BIOFISH parameters, three clusters were determined. Figure S2: (a) Scree plot depicting the eigenvalues of PCs obtained from PCA performed on BIOFISH parameters. Only two of the PCs have an eigenvalue greater than

1. (b) Variance explained by the designated factors. The first two factors explain a large part of the variance. (c) Rotated factor pattern for the six BIOFISH parameters used for PCA.

**Author Contributions:** Conceptualization, A.W., N.B., A.H. and S.N.; Data curation, A.W.; Formal analysis, A.W.; Funding acquisition, S.N.; Investigation, A.W., N.B. and J.Y.; Methodology, A.W., N.B., J.Y. and A.H.; Project administration, S.N.; Resources, A.H. and S.N.; Software, A.W. and S.N.; Supervision, N.B., A.H. and S.N.; Validation, N.B., A.H. and S.N.; Visualization, A.W.; Writing—original draft, A.W.; Writing—review and editing, A.W., N.B., J.Y., A.H. and S.N. All authors have read and agreed to the published version of the manuscript.

**Funding:** The research was funded by the Federal Ministry of Education and Research of Germany (BMBF, grant no. 02WQ1375A). Further, we acknowledge support by the KIT-Publication Fund of the Karlsruhe Institute of Technology for funding the publication.

**Institutional Review Board Statement:** Not applicable.

**Informed Consent Statement:** Not applicable.

**Data Availability Statement:** Not applicable.

**Conflicts of Interest:** The authors declare no conflict of interest.

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
