# Peer review of "Insights into Phytoplankton Dynamics and Water Quality Monitoring with the BIOFISH at the Elbe River, Germany"

_water, doi:10.3390/w14132078_

Round 1

Reviewer 1 Report

This study obtained some basic data and discharge data from CDC of the German and the FIS of the FGG Elbe, respectively. Also, some monitoring data of the rivers were also get using of an in-situ online multi-sensor-system like the BIOFISH device. Authors investigated sources of water quality variations and assessed the influence of phytoplankton dynamics, tributaries, and other pollution sources using different statistical methods, such as CA, PCA. On the whole, it is a well written and informative paper. But I think that most of the diagrams should be readjusted.

1)     I suggested to merge Figure.1 and Figure.2, as there was not much information in Figure.2.

2)     Figure 3: The icon should be placed on the coordinate axis.

3) Figure 4 and 5: I don't know what the y-coordinate means.

4) Figure 7: Correlation coefficients should be plotted on the graph.

5) Figure about PCA should also be shown in the paper or in SI.

Author Response

Dear Reviewer,

thank you for your effort in evaluating our manuscript. We are glad that you found our work worthwhile and well fitting for the scope of the Water MDPI journal. The suggestions/recommendations for the illustrations are profound and we will change them accordingly.

Moreover, although we are not completely sure which information about the PCA is missing, we will put the scree plot and the rotated factor pattern for the PCA in the supplementary material to give further insights to the results of the PCA. We hope this is what you thought is missing. If not, we will gladly put more detailed results of the PCA in the SI.

Many greetings and kind regards,

Andre Wilhelms

Reviewer 2 Report

This is an excellent descriptive paper outlining the results of the water quality parameters survey carried out in River Elbe.  This is a very preliminary type of study on the water quality parameters to describe and phytoplankton dynamics (in terms of phytoplankton biomass or Chl a) carried out on a very long stretch of River Elbe. The study describes very clearly the importance of the research, background, study sites, methodology, analyses protocols, results and discussion. The data were collected systematically and analysed using appropriate stat tests, and the results are presented and interpreted in an adequate way. The conclusion of the study provides the salient features and take-home messages clearly. Nothing fancy or complicated here, very straight forward report. At some places it looked like a report, not as a research paper, especially when the background and data collection were described. This makes the paper pretty long and descriptive. If the journal does not have any problem with the length and the style of writing, I have no hesitation to recommend acceptance of this paper. At some places I could see some redundancies, especially with respect to the description of the sites, sampling collection using the research vessel etc.. If this can be avoided and the length of the paper can be reduced, that would be awesome.

I could see in the paper the use of “chapter 4.1” (in discussion) which I think the authors refer to the results section, right? If so please change these terminologies. This is applied generally to reports or thesis.

Author Response

Dear Reviewer,

thank you for your effort in evaluating our manuscript. We are glad that you found our work worthwhile and well fitting for the scope of the Water MDPI journal. The suggestions/recommendations for the reduction of the background & methods sections are profound and we will (try to) reduce these segments as much as possible to get rid of unnecessary information. In generell your suggestions will definitely make our manuscript more wholesome.

Further, the hint about the links to different chapters in the discussion part was also a good suggestions. I already changed them/deleted them.

Many greetings and kind regards,

Andre Wilhelms